# *S-SOLVER*: Numerically stable adaptive step size solver for neural ODEs

## Abstract

A neural ordinary differential equation (ODE) is a relation between an unknown function and its derivatives, where the ODE is parameterized by a neural network. Therefore, to obtain a solution to a neural ODE requires a solver that performs numerical integration. Dopri5 is one of the most popular neural ODE solvers and also the default solver in *torchdiffeq*, a PyTorch library of ODE solvers. It is an adaptive step size solver based on the Runge-Kutta (RK) numerical methods. These methods rely on estimation of the local truncation error to select and adjust integration step size, which determines the numerical stability of the solution. A step size that is too large leads to numerical instability, while a step size that is too small may cause the solver to take unnecessarily many steps, which is computationally expensive and may even cause rounding error build up. Therefore, accurate local truncation error estimation is paramount for choosing an appropriate step size to obtain an accurate, numerically stable, and fast solution to the ODE. In this paper we propose a novel local truncation error approximation that is the first to consider solutions of four different RK orders to obtain a more reliable error estimate. This leads to a novel solver *S-SOLVER* (Stable Solver), which is more numerically stable; and therefore accurate. We demonstrate *S-SOLVER*'s competitive performance in experiments on image recognition with ODE-Net, learning hamiltonian dynamics with Symplectic ODE-Net, and continuous normalizing flows (CNF).

## 1 Introduction

Neural ODEs are continuous depth deep learning models that combine neural networks and ODEs. Since their first introduction in (Chen et al., 2018), they have been used in many applications such as: stochastic differential equations (Li et al., 2020), physically informed modeling (Sanchez-Gonzalez et al., 2019; Zhong et al., 2020), free-form continuous generative models (Grathwohl et al., 2019; Finlay et al., 2020), mean-field games (Ruthotto et al., 2020), and irregularly sampled time-series (Rubanova et al., 2019).

Neural ODEs parameterize the derivative of the hidden state using a neural network; and therefore, learn non-linear mappings via differential equations. A differential equation is a relation between an unknown function and its derivatives. Ordinary differential equations describe the change of only one variable (as opposed to multiple) with respect to time, i.e.: $dx/dt = f(t, x)$. Typically, an ODE is formulated as an initial value problem (IVP), which has the following form. Given a function derivative $dx/dt$, a time interval $t = (a, b)$ and an initial value (e.i.: $x$ at time $t = a$), the solution to the IVP yields $x$ evaluated at time $t = b$. The method for approximating $x(b)$ is numerical integration; therefore, all the various ODE solvers include different methods for performing integration.

Adaptive step size solvers are amongst the most popular solvers for neural ODEs. In fact, the default solver in *torchdiffeq* (a library of ODE solvers implemented in PyTorch) is Dopri5, the Dormand-Prince 5(4) embedded adaptive step size method of the Runge-Kutta (RK) family. Adaptive step size RK solvers perform two approximations: one of order $p$ and another of $p-1$ and compare them to obtain the local truncation error, which is used to determine the integration step size. Specifically, the error is used to make a decision whether to accept or reject the solution step under the current step size and to decide how to modify the step size for the next step. A step size that is too large leads

to numerical instability, while a step size that is too small may cause the solver to take unnecessarily many steps, which is computationally expensive and may even cause the rounding error to build up. Therefore, accurate local estimation is paramount for choosing an appropriate step size to obtain an accurate, numerically stable, and fast solution to the ODE.

The local truncation error is defined as the difference between the exact and approximate solution obtained at a given time step. All currently available adaptive step neural ODE solvers rely on estimating the local error as the difference between order $p$ and $p-1$ solutions, which assumes that the order $p$ solution is exact. This is not necessarily true and if the $p$ solution is far from the exact one, the local error estimate is inaccurate, which results in the solver making poor decisions regarding its step size.

In this paper we propose a novel local truncation error estimation that takes into account multiple orders of the RK method as opposed to just order $p$ and $p-1$ to obtain a more accurate estimate of the local truncation error that guides the integration step size. Specifically, we modify the local truncation error estimation of Dopri8, the Dormand-Prince 8(7) embedded adaptive step size method. Dopri8 calculates the local truncation error as the difference between its 8th and 7th order solution. Our modification computes this error as the average of the difference between both its 8th and 7th, and also 4th and 5th order solution. This leads to a new ODE solver, *S-SOLVER* (Stable Solver), a modified Dopri8 integrator with more accurate local truncation error estimation that provides more reliable information for step size calculations; and therefore, more numerically stable solution. To our best knowledge, *S-SOLVER* is the first solver that uses a multiple solution orders to estimate local truncation error for adjusting its step size.

## 2 BACKGROUND

### 2.1 NEURAL ORDINARY DIFFERENTIAL EQUATIONS

Traditional neural networks are defined as discrete models with a discrete sequence of hidden layers, where the depth of the network corresponds to the number of layers. Neural ODEs (Chen et al., 2018) are continuous depth deep learning models, which parameterize the derivative of the hidden state using a neural network. Specifically, they are ODEs that are parameterized by a neural network, which has many benefits such as memory efficiency, adaptive computation, and parameter efficiency.

Neural ODEs are inspired by the dynamic systems interpretation of residual and other networks Haber et al. (2018); Weinan (2017). These networks perform a sequence of transformations to a hidden state:

$$state_{t+1} = state_t + f(state_t, \theta_t), \tag{1}$$

which can be viewed as discretized forward Euler method applied to a continuous transformation. Given this interpretation, the transformation to a hidden state can be formulated as an ODE:

$$d\,state(t)/dt = f(state(t), t, \theta), \tag{2}$$

where $state(t = 0)$ is the input layer and $state(t = T)$ is the output layer. Therefore, the neural ODE is an IVP:

$$dx(t)/dt = f(t, x(t), \theta), \ for\ t_0 \le t \le t_1, \ subject\ to\ x(t_0) = x_{t_0}, \tag{3}$$

where $f(.,.,\theta)$ is the deep neural network, $x_{t_0}$ is the input, and $x_{t_1}$ is the output.

Neural ODEs are trainable through loss minimization, but due to their continuous nature the optimization process is slightly different from classical discrete deep learning models. The forward pass solves the ODE with an ODE solver and the backward pass computes the gradients either by backpropagating through the ODE solver or with the adjoint method (Chen et al., 2018). In this work we focus on the forward pass, which outputs a solution to the ODE.

## 2.2 NEURAL ODE SOLVERS

Solving neural ODEs that we generalized in Equation 3 requires numerical integration that can be described as follows:

$$x(t_1) = x(t_0) + \int_{t_0}^{t_1} f(t, x(t), \theta)dt \tag{4}$$

This equation can be solved with an ODE solver, which returns the value of $x(t_1)$ that represents the solution at the end of the time interval that satisfies the initial condition $x(t_0) = x_{t_0}$.

There are different types of ODE solvers that use different methods and algorithms for performing numerical integration, and the Runge-Kutta (RK) set of methods that are amongst the most popular (Seiler & Seiler, 1989). The basic idea behind the RK integration methods is to re-write $dx$ and $dt$ in Equation 3 as finite steps $\Delta x$ and $\Delta t$ and multiply the equations by $\Delta t$, which provides a change in $x$ with respect to $\Delta t$. The finite time step $\Delta t$ is called the step size (Seiler & Seiler, 1989) and is typically represented as $h$. The the simplest RK method, the Euler method, illustrates this well:

$$x_{t_{n+1}} = x_{t_n} + hf(t_n, x_{t_n}) \tag{5}$$

RK methods leverage the differential equation for computing the slope $k$ of the tangent line to the function $f$. The slope is then used to approximate $f$ at the next time step $t+1$. As shown in (Bogacki & Shampine, 1989), this can be represented as:

$$x_{t_{n+1}} = x_{t_n} + h\sum_{i=1}^{S} \hat{b}_i k_i, \tag{6}$$

where

$$
\begin{aligned}
k_1 &= f(t_n, \hat{x}_{t_n}) \\
k_i &= f(t_n + c_i h, \hat{x}_{t_n} + h\sum_{j=1}^{i-1} a_{ij}k_j) \; for \; RK \; stages \; i = 2, ..., s \\
c_i &= \sum_{j=1}^{i-1} a_{ij}
\end{aligned}
\tag{7}
$$

Since the Euler method approximates the slope only once to proceed from $t$ to $t+1$, it can be expressed using the general RK method shown in Equation 6 as:

$$x_{t_{n+1}} = x_{t_n} + h(b_1 k_1) \tag{8}$$

The number of times that the slope $k$ is approximated between $t$ and $t+1$ impacts the local truncation error the RK method Burden et al. (2015); Burrage & Burrage (2000). The local truncation error is the difference between the exact and approximated solution and determines the order of the RK method Burden et al. (2015). The order of the RK method corresponds to the order of the local truncation error minus one. For example, the local truncation error for the Euler's method is $O(h^2)$, resulting in a first order numerical technique.

## 3 NUMERICAL STABILITY OF NEURAL ODE SOLVERS

When approximating the solution of an IVP, there are two primary sources of error: the roundoff error and the truncation error (Abell & Braselton, 2014), which impact the numerical stability of the ODE solver that yields the approximate solution.

### 3.1 NUMERICAL STABILITY

Numerical stability can be viewed property of of an algorithm, which describes the sensitivity of a solution to numerical errors (Higham, 2002). An unstable numerical method produces large changes

in outputs in response to small changes in inputs (Jong, 1977), which can lead to unexpected outputs or errors. Numerical instability arises due rounding, and truncation errors (Higham, 2002). Roundoff errors are caused by approximating real numbers with finite precision, while truncation errors are caused by approximating a mathematical process. Many numerical methods (e.g.: Euler's method for solving differential equations) can be derived by taking finitely many terms of a Taylor series. The terms omitted constitute the truncation error, which often depends on a parameter called the step size (Higham, 2002). In this paper introduce a novel truncation error estimation used for setting an adaptive step size that achieves a numerically stable solution.

## 3.2 STABILITY OF DIFFERENT ODE SOLVERS

There are two notions of numerical stability of ODEs: zero-stability and absolute stability. Zero-stability implies that on a fixed time interval, small perturbations of data yield bounded perturbations in the solution as the step size h approaches zero (LeVeque, 2005). Absolute stability, a stronger notion of stability, guarantees the same behavior, but for a fixed step size h as the time interval approaches infinity. Generally, a numerical method for for solving initial value ODE is numerically stable if "small changes or perturbations in the initial conditions produce correspondingly small changes in the subsequent approximations" (Burden et al., 2015).

Different ODE solvers have different numerical stability. This can be demonstrated with a canonical example of an ODE that describes a swinging pendulum:

$$ml\frac{d^2\Theta(t)}{dt^2} = -mg\sin(\Theta(t)),\tag{9}$$

where $\Theta(t)$ is the angle between the pendulum and a vertical axis at a time $t$, $l$ is the length of the pendulum, $m$ is the pendulum mass, and $g$ represents gravity. Figure 1 illustrates the varying degrees of numerical stability of three different methods that can be used to solve this pendulum IVP.

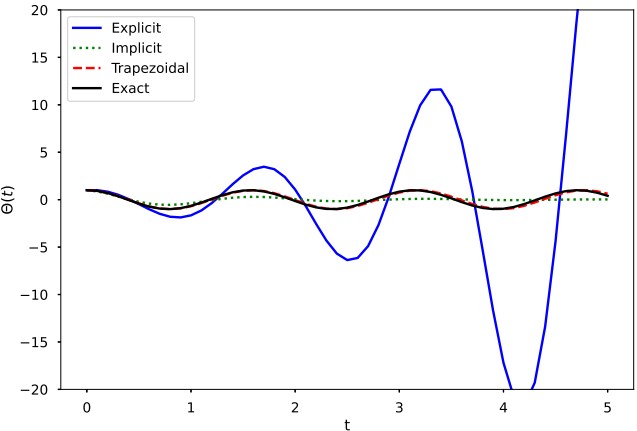

Figure 1: Comparison of numerical stability of various ODE solvers

## 3.3 ANALYSIS ON NUMERICAL STABILITY WITH RESPECT TO ODE SOLVER STEP SIZE

To illustrate the impact of step size on the numerical stability of ODE solvers, we provide a numerical stability analysis of the Euler method and derive its stability condition. The stability condition pertaining to explicit Euler method's step size can be derived using the *test equation*:

$$y' = ky, \quad y(0) = \alpha, \quad \alpha < 0\tag{10}$$

Applying the forward Euler method to this ODE yields:

$$x_0 = \alpha, \quad x_{i+1} = x_i + h(kx_i) = (1 + hk)x_i\tag{11}$$

Solving for $x_{i+1}$:

$$x_{i+1} = (1 + hk)x_i = (1 + hk)^{i+1}x_0 = (1 + hk)^{i+1}\alpha \tag{12}$$

The exact solution is:

$$y(t) = \alpha \exp(kt) \tag{13}$$

The absolute error is the absolute difference between the exact and approximated solution:

$$|y(t_i) - x_i| = |\exp(ihk) - (1 + hk)^i||\alpha| = |\exp(hk)^i - (1 + hk)^i||\alpha| \tag{14}$$

If $k > 0$, the problem is unstable. If $k \le 0$ and $|1 + hk| < 1$, the forward Euler method will be stable. This condition is called the *stability region* and pertains to the notion of absolute stability. Specifically, the analysis of the *stability region* is useful for determining a step size that can ensure absolute stability.

### 3.3.1 ILLUSTRATIVE EXAMPLE

We demonstrate the practical application of the theoretical numerical stability analysis shown above with an illustrative example of an ODE $dy/dt = -2.3y$ with an initial value of $y(0) = 1$. Figure 2a compares the exact solution $-2.3t$ with approximate solutions obtained with the explicit Euler method with varying step sizes: $h = 1.0, 0.7, 0.1$. The solution obtained with step size $h = 1.0$ is erratic and inaccurate, while the solution with the smallest step size $h = 0.1$ yields a stable solution that is very close to the exact one. The reason for that is that $kh$ for $h = 1.0$ and $h = 0.7$ are far away from the *stability region* represented as the blue circle in Figure 2b. Therefore, we can observe that the step size $h$ has a significant impact on the numerical stability and accuracy of the ODE solution.

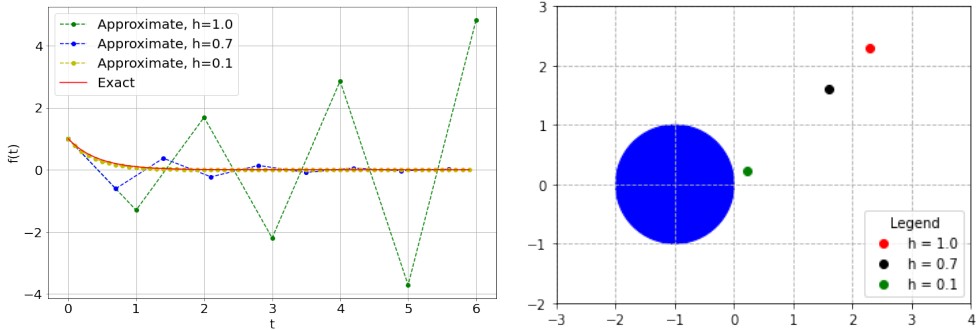

(a) The importance of step size for accurate solutions  (b) The stability region of the explicit Euler method

Figure 2: The relationship between numerical stability of an ODE solution and ODE solver step size

## 4 METHOD

Prior adaptive step size solvers approximate the local truncation error as the difference between order $p - 1$ and $p$ solution, where the $p$ order solution is assumed to be the exact solution. This means that the error estimates are not exact, but only accurate to the leading order in $h$, i.e.: order $p$ (Press & Teukolsky, 1992). *S-SOLVER* is an adaptive step size solver with novel, more accurate local error estimation that is used for adjusting the solver step size to achieve an accurate numerically stable solution to neural ODEs.

*S-SOLVER* is based on Dopri8, the Dormand-Prince 8(7) embedded adaptive step size method, which is a 8th order RK method that requires 13 function evaluations per integration step (Prince & Dormand, 1981) as shown in Equation 15.

$$x_{t_{n+1}} = x_{t_n} + h \sum_{i=1}^{13} \hat{b}_i k_i, \tag{15}$$

where

$$k_1 = f(t_n, \hat{x}_{t_n})$$

$$k_i = f(t_n + c_i h, \hat{x}_{t_n} + h \sum_{j=1}^{i-1} a_{ij} k_j) \ for \ i = 2, ..., 12 \tag{16}$$

$$c_i = \sum_{j=1}^{i-1} a_{ij}$$

The coefficients a, b, and c in Equations 15 and 16 are defined using the Butcher tableau provided in (Prince & Dormand, 1981).

In contrast to Dopri8, which calculates local error as the difference between the 7th and 8th order solution, *S-SOLVER* uses order 8, 7, 5, and 4. Specifically, given a neural ODE:

$$x(t_1) = x(t_0) + \int_{t_0}^{t_1} f(t, x(t), \theta) dt, \tag{17}$$

suppose that the solver has progressed in integration to some time step $t_n$ and approximated $x(t_n)$ as $\hat{x}(t_n)$. To make further progress, the solver needs to take a step forward and compute the value of $x$ at time step $t_n + h$, where $h$ is the step size. Suppose that this is approximated as $\hat{x}(t_n + h)$ and that the step's error is $x_{error}$. *S-SOLVER* computes $x_{error}$ as an average of the difference between 8th and 7th order solution and 5th and 4th order solution to obtain a more reliable estimate:

$$x_{error} = \frac{(\hat{x}(t_n + h)_{order_8} - \hat{x}(t_n + h)_{order_7}) + (\hat{x}(t_n + h)_{order_5} - \hat{x}(t_n + h)_{order_4})}{2} \tag{18}$$

The 5th and 4th order solution is computed using a similar process, but only with 6 stages as follows:

$$x_{t_{n+1}} = x_{t_n} + h \sum_{i=1}^{6} \hat{b}_i k_i, \tag{19}$$

where

$$k_1 = f(t_n, \hat{x}_{t_n})$$

$$k_i = f(t_n + c_i h, \hat{x}_{t_n} + h \sum_{j=1}^{i-1} a_{ij} k_j) \ for \ i = 2, ..., 5 \tag{20}$$

$$c_i = \sum_{j=1}^{i-1} a_{ij},$$

where the coefficients a, b, and c are given in the Butcher tableau provided in (Lawrence, 1986).

Given a pre-defined upper bound on relative error RTOL (1e-7 default in *torchdiffeq*) and upper bound on absolute error ATOL (1e-9 default in *torchdiff*), the solver then computes an error ratio $r$ as follows:

$$r = \|\frac{x_{error}}{scale}\|, \tag{21}$$

where *scale* is defined as:

$$scale = ATOL + RTOL \ \max(\hat{x}(t_n), \hat{x}(t_n + h)). \tag{22}$$

If $r \leq 1$ the step is accepted, otherwise it is rejected and the value of $x$ at time step $t_n + h$ is approximated again with a smaller step size $h$.

## 5 EXPERIMENTS

We implement *S-SOLVER* as a new solver that is part of the torchdiffeq library (`https://anonymous.4open.science/r/S-SOLVER-EC78/ReadMe.md`) and perform experiments on image recognition with ODE-Net, learning hamiltonian dynamics with Symplectic ODE-Net, and generating new distributions with continuous normalizing flows (CNF). We demonstrate the *S-SOLVER* is accurate and numerically stable thanks to better local error estimation that determines the step size, which in turn afects the numerical stability of the ODE solution as shown in section 3.

### 5.1 STIFF NEURAL ODE AND ERROR MONITORING

We first validate *S-SOLVER*'s numerical stability on solving the following stiff neural ODE obtained from page 353 of Burden et al. (2015):

$$dy/dt = 5\exp(5t)(y-t)^2 + 1 \; for \; 0 \le t \le 1, \; subject \; to \; y(t=0) = -1 \tag{23}$$

This ODE equation is stiff, which means that it is likely the error due to approximation is amplified and becomes dominating in the solution calculations leading to a numerically unstable solution (Burden et al., 2015; Kim et al., 2021).

As shown in Figure 3, the neural ODE solved with *S-SOLVER* yields a solution that is very close to the exact solution:

$$y(t) = t - exp(-5t) \tag{24}$$

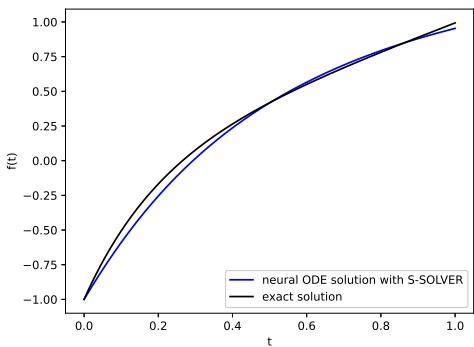

Figure 3: Solution to a stiff Neural ODE with *S-SOLVER*

In addition to demonstrating that *S-SOLVER* can solve stiff neural ODEs, which typically have numerical stability issues, we also examine the local error. Figure 4 shows the local error estimate produced by Dopri5 (default solver in *torchdiffeq*), *S-SOLVER*, and also a comparison of the two, which suggests that Dopri5 underestimates the local error.

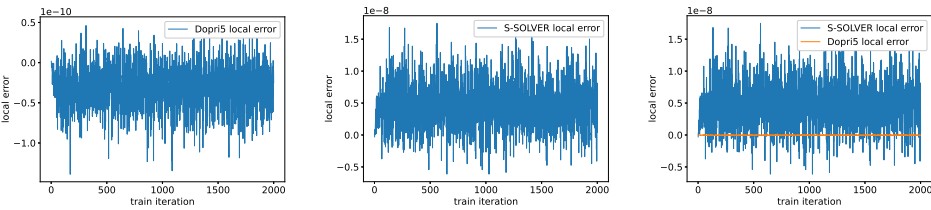

Figure 4: A comparison of the local error produced by Dopri5 and *S-SOLVER*

## 5.2 IMAGE RECOGNITION

The next set of experiments focuses on image recognition with ODE-Nets. We train an ODE-Net with *S-SOLVER* and compare its results with an ODE-Net trained with Dopri5 (default solver in *torchdiffeq*) and also a classical ResNet on two datasets: MNIST and FASHION MNIST. Table 1 shows that the highest test accuracy on both datasets is achieved with our ODE-Net with *S-SOLVER*. The test accuracy on MNIST beats prior SOTA results in (Chen et al., 2018), who report a 0.42% test error, i.e.: 99.58% test accuracy. Using the same experiment settings as (Chen et al., 2018), thanks to *S-SOLVER* we push the test accuracy to 99.73%. Our results are also better compared to, for example, (Ghosh et al., 2020) who report 98.3% test accuracy that is achieved with their proposed temporal regularization.

Table 1: Results for ODE-Net with *S-SOLVER* on image recognition tasks

|  | MNIST | | | FASHION MNIST | | |
| --- | --- | --- | --- | --- | --- | --- |
|  | train acc | test acc | loss | train acc | test acc | loss |
| **ODE-Net with *S-SOLVER*** | 99.99% | **99.73%** | 0.00013 | 97.58% | **94.00%** | 0.079816 |
| ODE-Net with dopri5 | 99.98% | 99.69% | 0.04623 | 97.75% | 93.72% | 0.055531 |
| ResNet | 99.96% | 99.68% | 7.2E-05 | 98.52% | 93.94% | 0.065489 |

## 5.3 LEARNING HAMILTONIAN DYNAMICS

We test *S-SOLVER* on Symplectic ODE-Net (Zhong et al., 2020), which can learn Hamiltonian dynamics. Specifically, we choose the problem of "acrobot" (Murray & Hauser, 2010; Sutton & Barto, 2005), which simulates a physical system with two joints and two links, where the joint between the two links is actuated. Initially, the links are hanging downwards, and the goal is to swing the end of the lower link up to a given height. In Figure 5 we show that the validation loss obtained with *S-SOLVER* is more stable than with Dopri5 (default solver in torchdiffeq) and therefore, preferable. We interpret this observation to be the result of *S-SOLVER*'s more reliable local estimation that controls the step size, which in turn impacts the stability of the ODE solution.

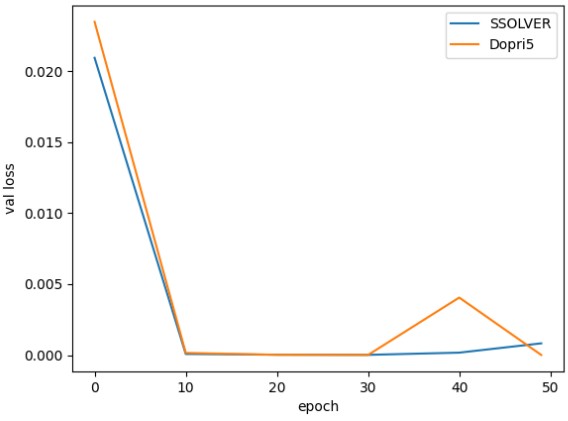

Figure 5: Acrobot: The validation loss obtained by solving Symplectic ODE-Net with *S-SOLVER* is more stable than with Dopri5 solver (default solver in *torchdiffeq*)

## 5.4 CONTINUOUS NORMALIZING FLOWS

Continuous Normalizing Flows (CNF) are generative models introduced by (Chen et al., 2018) that leverage neural ODEs. CNFs are based on normalizing flows (Rezende & Mohamed, 2015), which perform transformations of a simple probability distribution into a more complex one by a sequence of invertible and differentiable mappings (Kobyzev et al., 2021).

We perform experiments with CNFs that use *S-SOLVER* and visualize how the model generates the Two Circles distribution from random noise in Figure 6. Figure 6 shows the evolution of the generated distribution (samples) and probability density (log probability) with respect to the Two Circles distribution (target) over time from time-step 0.0 to 10.0. It can be observed that by the last time step, the random distribution has been transformed into the Two Circles distribution.

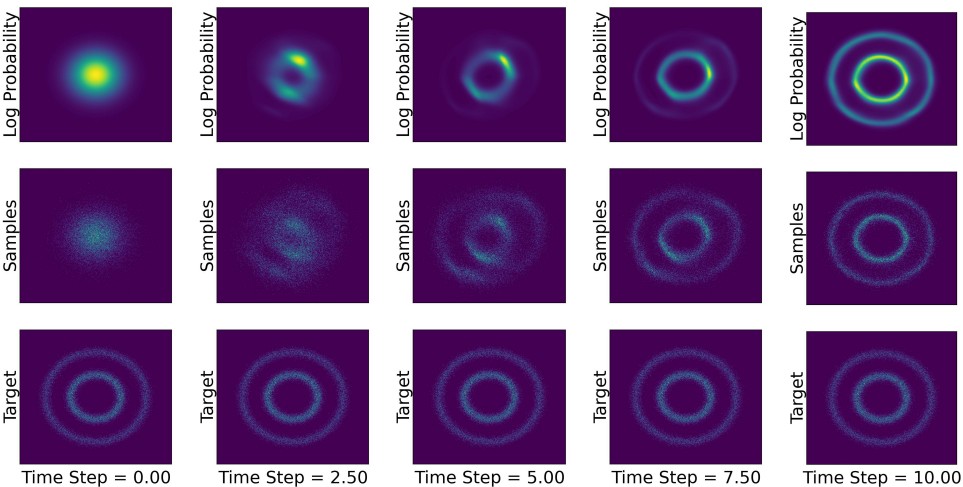

Figure 6: Continuous normalizing flows for fitting the Two Circles distribution with *S-SOLVER*

## 6    RELATED WORK

This paper focuses on ODE solvers for solving neural ODEs, which involves performing numerical integration. Specifically, we focus on adaptive step size ODE solvers which have become the standard for solving neural ODEs. While to our best knowledge, we are the first ones to propose a more numerically stable ODE solver that is based on more accurate local truncation error estimation, there are several prior works that also study numerical integration in neural ODEs. Zhu et al. (2022) perform numerical analysis of numerical integration in neural ODEs and propose IMDE, or inverse modified differential equations. Zhuang et al. (2021) propose MALI, a new numerical integrator that is memory-efficient. Ghosh et al. (2020) introduce STEER, a simple temporal regularization that randomly perturbs the numerical integration time limits. Pal et al. (2021) propose a regularization method for adaptive ODE solvers that uses the internal cost heuristics. Yan et al. (2020) study the robustness of the Euler method, which is the simplest, but important neural ODE solver. Krishnapriyan et al. (2022) develop a convergence test that can be used to select an ODE solver that is suitable for a particular task.

## 7    CONCLUSION

In this paper we demonstrate the importance of appropriately choosing and adapting the step size in ODE solvers for obtaining a numerically stable; and therefore, accurate solutions to a neural ODEs. To this end we propose *S-SOLVER*, a new neural ODE solver that is more numerically stable thanks to more accurate local truncation error estimation that is based on comparing multiple approximations as opposed to just two, which has been the standard approach. We provide a theoretical analysis of the impact of solver step size on numerical stability and also perform practical experiments with *S-SOLVER*. We show that *S-SOLVER* can solve a stiff neural ODE and that image recognition ODE-Nets learned with *S-SOLVER* surpass the test accuracy of prior solvers as well as classical ResNets on MNIST and FASHION MNIST. In fact, *S-SOLVER* achieves a new SOTA test accuracy on MNIST. We also show that the process of learning Hamiltonian dynamics with Symplectic ODE-Nets on the acrobot example is more stable with *S-SOLVER* than with Dopri5, the solver used in prior neural ODE works. Finally, we also show that *S-SOLVER* works well for CNFs in an experiment, where we successfully learn a new data distribution from random noise.

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
