# OpenReview forum: "S-SOLVER: Numerically Stable Adaptive Step Size Solver for Neural ODEs"
_ICLR.cc/2023/Conference — Submitted to ICLR 2023_

### Official Review · Reviewer_cFyQ · 2022-10-19

**Confidence:** 4
**Correctness:** 2
**Technical Novelty And Significance:** 1
**Empirical Novelty And Significance:** Not applicable
**Recommendation:** 1

**Clarity, Quality, Novelty And Reproducibility:**

The novelty is unclear.

The coefficients are unclear, so the reproducibility is limited.

**Strength And Weaknesses:**

**Strength**

It worked better than dopri5 for some tasks of neural ODEs.

**Weakness**

The main concern is whether the proposed method is suited for ICLR publication. The proposal is a higher-order adaptive stepping numerical solver, which is generally applicable to any ODEs and not specialized to neural ODEs. I think the present study is about numerical analysis and is out of scope of ICLR.

There is no theoretical support that the error estimate in (8) is more accurate than that of dopri8. It appears that (8) is merely an average of the estimates by dopri8 and dopri5.

S-Solver uses the methods of orders 8,7,5,4 to estimate the local truncation error. The computation cost is simply increased from dopri8 or dopri5.

The coefficients of Runge-Kutta methods are not uniquely determined, but they are not provided explicitly. They are the same as those used for dopri8 and dopri5?

For any adaptive stepping methods, one can choose the tolerance to obtain arbitrary accuracy. So, it is unclear why the proposed method worked better than dopri5. Because the proposed method is based on dopri8, it should be compared with dopri8, not dopri5.

**Summary Of The Paper:**

This study proposed S-SOLVER, which is a numerical integrator with adaptive step size based on eighth-order Dormand-Prince method. It can solve even a stiff equation.


**Summary Of The Review:**

The proposed method appears to be just a mixture of dopri5 and dopri8, with no theoretical support that it is superior to dopri8. In addition, the proposed method is related to numerical analysis and not directly to learning. Hence, I do not feel the need to publish it from ICLR.

---

### Official Review · Reviewer_H6ea · 2022-10-22

**Confidence:** 5
**Correctness:** 1
**Technical Novelty And Significance:** 1
**Empirical Novelty And Significance:** 1
**Recommendation:** 1

**Clarity, Quality, Novelty And Reproducibility:**

Clarity: Overall, the paper is easy to read and the context presented in this paper is quite simple, which is basically on some basic numerical ODE solvers.

Quality: In my opinion, the quality of this paper is low. The proposed s-solver is not more accurate than dopri5 but computationally much more expensive. There have been existing local error estimations using RK methods of different orders developed several decades ago, which were ignored in this paper.

Reproducibility: No details of the experimental setting are provided in the paper.

**Details Of Ethics Concerns:**

Not available

**Strength And Weaknesses:**

Strength: the paper is easy to read, and the context is most about basic numerical ODE solvers.

Weakness: I have many concerns about this work. In particular, the novelty, accuracy, and computational efficiency of the proposed method compared to the standard adaptive ODE solver. Below, I list some detailed comments.

1. The contribution of this paper is very marginal and seems problematic to me. As stated in the paper about the contribution of this paper "We modify the local truncation error estimation of Dopri8, the Dormand-Prince 8(7) embedded adaptive step size method. Dopri8 calculates the local truncation error as the difference between its 8th and 7th-order solution. Our modification computes this error as the average of the difference between both its 8th and 7th, and also 4th and 5th order solution." --- This will make the resulting ODE solver has no better convergence order but computationally much more expensive than Dopri5.  In particular, in each ODE numerical integration, we need to evaluate the right hand side of the Neural ODE many more times than the standard methods that have the same order of accuracy at best.


2. Many existing methods, which have been developed for several decades, use multiple different-order methods for local truncation error estimation. The authors may refer to the book “Hairer, E., Nørsett, S.P., Wanner, G., Solving Ordinary Differential Equations I, Nonstiff Problems, Second Revised Edition, Springer, 2008” and “Hairer, E., Wanner, G., Solving Ordinary Differential Equations II, Stiff and Differential-Algebraic Problems, Second Revised Edition, Springer, 2002”. As far as I remember, DOP853 belongs to such category.


3. The authors seem confused about numerical stability and numerical accuracy. I do not see that s-solver is more stable. Usually higher-order explicit ODE solvers are less stable than the low-order explicit methods. Implicit ODE solvers are much more stable than explicit ODE solvers.


4. The ODE in section 5.1 is not a stiff ODE. The dynamics of a stiff ODE should change abruptly. The following tutorial provides you an example of stiff ODE
https://www.mathworks.com/company/newsletters/articles/stiff-differential-equations.html

Again, this example shows that the authors seem to have some misunderstanding of stiff ODEs.


5. Figure 3 is meaningless; you should contrast s-solver with other ODE solvers. Also, y-label should be y(t) rather than f(t). Why the error between the numerical solution and the exact one is so large when t is around 0.2? I also want to see the behavior of numerical and exact solutions when t is much larger than 1. For instance, you may plot for t from 0 to 10.

6. All currently available adaptive step neural ODE solvers rely on estimating the local error as the difference between order $p$ and $p − 1$ solutions, which assumes that the order $p$ solution is exact. This is not necessarily true, and if the $p$ solution is far from the exact one, the local error estimate is inaccurate, which results in the solver making poor decisions regarding its step size. - - - This statement seems wrong.


7. The authors claim s-solver is stable; why is this the case? I would like to see some discuss of this. For instance, the authors may follow some existing discussions on the stability of ODE solvers. E.g., https://math.stackexchange.com/questions/2282814/stability-of-numerical-ode-solvers


8. There are many issues with the experiments: 1) There is no detail of the experimental settings, e.g., hyperparameters from training the machine learning model, neural network architecture, etc. 2) The testing examples are rather toyed examples. 3) The authors did not report the computational time.


9. The authors may test the impact of the error of dopri5 and s-solver on the performance of the trained machine learning model.



Minor comments:

1. The writing style is really not appreciated. The entire section 2, which takes slightly less than two pages, is reviewing existing neural ODE. The entire section 3, which takes around two pages, is simply adapted from some standard numerical ODE textbook. The paper contains really minimal new knowledge.

2. There are some typos. E.g., in the abstract, "build up" should be replaced with "build-up".


**Summary Of The Paper:**

The step size for the ODE solvers is crucial for training Neural ODEs. A step size that is too large leads to numerical instability, while a step size that is too small may cause the solver to take unnecessarily many steps, which is computationally expensive and may even cause rounding error build-up. Therefore, accurate local truncation error estimation is paramount for choosing an appropriate step size to obtain an accurate, numerically stable, and fast solution to the ODE. In the paper under review, the authors propose a local truncation error approximation that is the first to consider solutions of four different RK orders to obtain a more reliable error estimate. This leads to a novel solver, S-SOLVER (Stable Solver), which is more numerically stable; and, therefore, accurate. The authors test S-SOLVER’s performance in experiments on image recognition with ODE-Net, learning hamiltonian dynamics with Symplectic ODE-Net, and continuous normalizing flows (CNF).

**Summary Of The Review:**

I feel sorry to provide a strong rejection for this paper as my concerns are substantial about this paper. First, the authors seem confused about the concept of numerical accuracy and numerical stability. Second, the proposed s-solver is not more accurate than the existing adaptive ODE solver but is computationally much more expensive. Third, the evaluation performed in this paper is very weak, and no details is provided in the experimental section to reproduce the numerical results.

---

### Official Review · Reviewer_BCpH · 2022-10-25

**Confidence:** 4
**Correctness:** 2
**Technical Novelty And Significance:** 2
**Empirical Novelty And Significance:** 2
**Recommendation:** 5

**Clarity, Quality, Novelty And Reproducibility:**

Quality:
Not very good quality. The proposed method is not well supported in theory (actually counter intuition). The experiments are also not extensive.

Clarity:
The paper is well-written and easy to follow.

Originality:
The idea is new to me (though looks not solid).

**Strength And Weaknesses:**

Strengths:
To my knowledge, solving stiff ODEs is not extensively studied in the field of Neural ODE. The authors are discussing an important yet not well studied problem.
Furthermore, the paper is in general well-written and easy to follow.

Weakness:
I have a few questions and hope the authors could address them, listed below.

1) On the theoretical groundings of the proposed method. I'm not sure if the authors have thought about this, but Runge-Kutta methods are derived by solving a system of equations, see wikipedia for detail https://en.wikipedia.org/wiki/Runge–Kutta_methods#Adaptive_Runge–Kutta_methods. The idea is that, a $p$-th order RK method should have a error on the order of $h^p$ where $h$ is the stepsize, and the system of equations are derived to satisfy this constraint.
Under this consideration, the 8-th and 7-th equations gives an error estimate of order $h^8$, and the difference between 5-th and 4-th equations give an error estimate of order $h^5$. I don't think it's meaning to average two errors of different orders.

2) On experiments. The experimental validations are focused on toy examples rather than some harder problems. For example, Neural ODE has been applied to ImageNet classification (1000 classes), CNF on Cifar10 and ImageNet64, which are much more complicated than the experiments on MNIST and Swiss roll here.

3) (Minor, some missing related works). There are some related works that the author could consider for reference, references [1] [2] consider the numerical issue with Neural ODE solvers and have achieved SOTA results on much harder experiments.

[1] Zhuang, Juntang, et al. "Adaptive checkpoint adjoint method for gradient estimation in neural ode." International Conference on Machine Learning. PMLR, 2020.
[2] Matsubara, Takashi, Yuto Miyatake, and Takaharu Yaguchi. "Symplectic adjoint method for exact gradient of neural ode with minimal memory." Advances in Neural Information Processing Systems 34 (2021): 20772-20784.

**Summary Of The Paper:**

The authors aim to propose a new ODE solver for Neural ODE, specifically targeted at stiff ODEs. To achieve this goal, the authors modified the Dopri5 method, and give an estimate of the local error as the mean of error estimate of 5-th order and 8-th order. The authors also validated their method in Neural ODE experiments.

**Summary Of The Review:**

The proposed method averages error estimates of different orders, which is not meaningful in theory. Furthermore, the experimental validations are too simple.

---

### Decision · Program_Chairs · 2023-01-20

**Decision:**

Reject

**Justification For Why Not Higher Score:**

My reason is articulated above.

**Justification For Why Not Lower Score:**

N/A

**Metareview: Summary, Strengths And Weaknesses:**

All three reviewers had major concerns with this paper regarding validity, suitability, and clarity. The authors did not engage in discussion. A clear reject.